# Nifedipine Improves the Ketogenic Diet Effect on Insulin-Resistance-Induced Cognitive Dysfunction in Rats

**DOI:** 10.3390/ph17081054

**Published:** 2024-08-10

**Authors:** Nancy M. Abdel-Kareem, Shimaa M. Elshazly, May A. Abd El Fattah, Afaf A. Aldahish, Sawsan A. Zaitone, Sahar K. Ali, Enas A. Abd El-Haleim

**Affiliations:** 1Department of Pharmacology and Toxicology, Faculty of Pharmacy, Sinai University—Arish Branch, Arish 45511, Egypt; 2Department of Pharmacology and Toxicology, Faculty of Pharmacy, Zagazig University, Zagazig 44519, Egypt; smelshazly@pharmacy.zu.edu.eg; 3Department of Pharmacology and Toxicology, Faculty of Pharmacy, Cairo University, Cairo 11562, Egyptenas.ahmed@pharma.cu.edu.eg (E.A.A.E.-H.); 4Department of Pharmacology and Toxicology, College of Pharmacy, King Khalid University, Abha 61441, Saudi Arabia; adahesh@kku.edu.sa; 5Department of Pharmacology and Toxicology, Faculty of Pharmacy, University of Tabuk, Tabuk 71491, Saudi Arabia; szaitone@ut.edu.sa; 6Department of Basic Medical Sciences, College of Medicine, AlMaarefa University, P.O. Box 71666, Riyadh 11597, Saudi Arabia; skamal@um.edu.sa

**Keywords:** nifedipine, insulin resistance, cognitive dysfunction, ketogenic diet, fructose

## Abstract

Insulin resistance, induced by high fructose consumption, affects cognitive function negatively. Nifedipine may be suggested for neurological disorders. This study aimed to assess the effect of nifedipine with either a normal diet (ND) or a ketogenic diet (KD) in cognitive dysfunction. Male Wistar rats received 10% fructose in drinking water for 8 weeks to induce insulin resistance. Rats received nifedipine (5.2 mg/kg/day; p.o.) later with ND or KD for an additional five weeks. One and two-way ANOVAs were used in analyzing the data. Reversion to the ND improved insulin resistance and lipid profile, besides brain-derived neurotrophic factor (BDNF), glycogen synthase kinase-3 beta (GSK3β), and insulin-degrading enzyme (IDE) levels. Rats fed KD alone and those that received nifedipine with KD did not show similar improvement in the previously mentioned parameters as the ND group. However, nifedipine-ND rats showed improvement in cognitive behavior and insulin resistance. Treatment with nifedipine-KD ameliorated GSK3β, amyloid β (Aβ), and tau protein levels. As the nifedipine-KD combination succeeded in diminishing the accumulated Aβ and tau protein, KD may be used for a while due to its side effects, then nifedipine treatment could be continued with an ND. This conclusion is based on the finding that this combination mitigated insulin resistance with the associated improved behavior.

## 1. Introduction

The blood–brain barrier (BBB) allows circulating insulin, released by Langerhans islet beta cells, to have a central influence in addition to peripheral effects, including regulating blood sugar levels, increasing glycogenesis, reducing lipolysis, and influencing inflammation [1]. It is believed that insulin is biosynthesized in the brain because of the high levels of insulin seen in the pons, medulla, and hypothalamus [2]. Insulin’s role in the brain extends its normal function as a glucose-regulating hormone to the regulation of different brain functions such as recognizing objects, processing positive emotions and rewards, integration of sensory information, and inhibitory control of eating and as a central regulator of whole-body energy and homeostasis [3]. Hyperinsulinemia, poor glucose metabolism, hyperlipidemia, and obesity are all signs of insulin resistance in metabolic syndrome [3,4]. It may manifest as a result of hereditary or environmental causes, such as stress, inactivity, and smoking [3]. Knowing that both insulin and amyloid-beta (Aβ) are broken down by the insulin-degrading enzyme (IDE), long-standing insulin resistance particularly centrally goes beyond the development of type 2 diabetes to impair neural and cognitive functions [2].

Despite its molecular composition that is comparable to that of glucose, fructose, a natural sugar component of consumed fruits and honey [5], does not raise blood sugar levels as much as other carbs do [6]. Evidence-based research confirms that poor eating habits and high-fructose diets cause cognitive impairment because the hippocampus, which is essential for learning and memorization, is very susceptible to high fructose levels [7,8]. Numerous studies have proposed several mechanisms explaining how excessive fructose consumption without restriction causes cognitive dysfunction by impairing mitochondrial function, raising levels of oxidative stress and inflammatory mediators, and reducing the expression of neurotrophic factors. These variables all interfere with cell metabolism and synaptic plasticity, which leads to neurological dysfunction [3].

The ketogenic diet’s recent widespread appeal stems from its capacity to lower body weight. It is prescribed as a complementary therapy in the case of refractory epilepsy, and its results in neurological illnesses were encouraging [9].

Nifedipine is an antihypertensive drug that belongs to dihydropyridine calcium channel blockers [10]. Cognitive dysfunction is characterized by increasing Ca^2+^ influx in neurons’ cytosols. Accumulation of Ca^2+^ inside neural cells results in deposition of Aβ and tau hyperphosphorylation, in addition to its negative effects on mitochondria. Hence, the usage of calcium channel blockers may help restore normal cognitive function [11].

The current study aims to explore the nifedipine therapeutic impact accompanied by conversion to a normal diet (ND) and ketogenic diet (KD) on rats with cognitive impairment associated with fructose-induced insulin resistance.

## 2. Results

### 2.1. Reversion to ND, with or without Nifedipine Therapy, Caused a Profound Body Weight Reduction

At the end of the experiment, after 13 weeks, there was a marked decrease in body weight of the insulin resistance (IR) + ND group, with and without nifedipine treatment, in addition to the IR + KD + nifedipine-treated group compared to the normal rat group. Meanwhile, IR + KD alone failed to reduce body weight compared to the normal rat group. There was a significant increase in body weight in the KD groups compared to the corresponding IR + ND groups (Figure 1 and Table 1).

Changes in body weight, according to the mean body weight after 8 weeks of treatment with 10% fructose, were assessed as (body weight at the end of 8th week − body weight at the start of the experiment). Then the mean final body weight after nifedipine and nutrition treatment for an additional 5 weeks, assessed as (body weight at the final day of the experiment – body weight at the end of 8th week). Then, we calculated the percentage by dividing by the start weight for each. ND: normal diet, KD: ketogenic diet.

### 2.2. Fructose (10%)-Administration-Induced Insulin Resistance

The effect of eight weeks of 10% fructose treatment on blood glucose levels was evaluated, to clarify the occurrence of insulin resistance, before glucose administration and at 30, 60, and 90 min after the 2.5 mg/kg of glucose administration. There was no significant increase at 0 and 30 min, but after 60 and 90 min, there was a significant increase in blood glucose levels in the 10%-fructose-treated groups compared to the non-treated group (Figure 2).

### 2.3. Only Reversion to ND Succeeded in Normalizing the HOMA-IR Index

The treatment effect on insulin resistance was evaluated by measuring the homeostatic model assessment of insulin resistance (HOMA-IR) index. HOMA-IR was calculated as follows: [FBG (mg/dL) × FINS (μIU/mL)/405] [12] (USA).

IR + KD, with or without nifedipine, failed to restore the HOMA-IR index to normal. A significant increase in fasting blood glucose (FBG), fasting insulin (FINS), and HOMA-IR index was noticed in the IR + KD group and both groups that received nifedipine compared to the normal rat group. Moreover, in IR + KD rats and those treated with nifedipine and fed with a KD, the HOMA-IR index was significantly higher compared to the IR + ND group. Interestingly, only reversion to ND succeeded in normalizing the HOMA-IR index (Table 2).

### 2.4. KD and Nifedipine-KD Improved Long-Term Memory in the Morris Water Maze (MWM) Behavioral Test

The acquisition phase was carried out in the MWM test to evaluate the treatment effect on short-term memory. Day 1: rats in both groups treated with nifedipine spent significantly more time finding the escape platform than the normal rat group. On days 2 and 3 there was no significant difference between groups in finding the platform. But, on the 4th day, IR+ KD + nifedipine showed a marked decrease in time spent in the water searching for the platform as compared to normal rats.

On the other hand, long-term memory was evaluated in the probe phase of the MWM. Interestingly, there was no significant difference between groups (Figure 3).

### 2.5. Lipid Profile: Cholesterol, Triglycerides (TGs), High-Density Lipoprotein (HDL), and Low-Density Lipoprotein (LDL) Were Measured to Estimate the Effect of Nifedipine and the Type of Nutrition on the Lipid Profile That Affects Cognitive Dysfunction

HDL level was significantly decreased in all groups compared to the normal rat group. IR + KD, with and without nifedipine, and IR + ND+ nifedipine significantly decreased HDL levels compared to the IR + ND group. Cholesterol, TGs, and LDL levels were significantly elevated in both ketogenic groups, with and without nifedipine, and the IR + ND + nifedipine group compared to the normal rats and IR + ND groups. However, nifedipine treatment with both diets, KD and ND, significantly decreased the LDL levels compared to the IR + KD group (Figure 4).

### 2.6. ND Normalized the Brain-Derived Neurotrophic Factor (BDNF) Level

BDNF level was measured to evaluate the effect of nifedipine and type of nutrition in enhancing cognitive function. Only reversion to the ND restored the BDNF to normal levels. On the other hand, a significant reduction in BDNF levels was observed in IR + KD, IR + ND + nifedipine, and IR + KD + nifedipine groups compared to the normal rat group (Figure 5).

### 2.7. Nifedipine and KD Failed to Improve Insulin Signaling and Metabolism

Again, ND succeeded in improving insulin signaling and metabolism, appeared to normalize the glycogen synthase kinase-3 beta (GSK3β) and IDE levels, and there was a significant increase in GSK3β levels in the IR + KD group and both groups treated with nifedipine compared to the normal rat group and IR + ND groups. However, both nifedipine groups failed to improve the GSK3β level to near the achieved level in the IR + KD group. In addition, GSK3β level in the IR + KD + nifedipine group was significantly reduced compared to the IR + ND + nifedipine group (Figure 6a).

IDE level was significantly reduced in the IR + KD group and both groups that were treated with nifedipine in contrast to the normal rats and the IR + ND groups (Figure 6b).

### 2.8. Nifedipine and KD Improved the Cerebral Cortex and Hippocampal Histopathological Score

The severity of insulin-resistance-induced cerebral cortex and hippocampus lesions was graded microscopically based on the degree and extent of tissue damage using a four-point scale: absent (grade 0), no lesions detected. Minimal (grade 1) lesions involved less than 15% of the tissue section. Mild (grade 2) lesions involved 15–45% of the tissue section. Moderate (grade 3) lesions involved 45–75% of the tissue section. Marked (grade 4) lesions involved greater than 75% of the tissue section. Neuronal tissues were examined for the following pathological changes: tissue edema, red neurons, neuronal pyknosis, perineuronal edema, neuronophagia of degenerated neurons, areas of necrosis, and areas of reactive gliosis.

Figure 7 shows photomicrographs of the cerebral cortex and hippocampus from several experimental groups. Normal rats showed preserved brain tissue histoarchitecture. Neuronal damage was seen in both brain tissues of study (cerebral cortex and hippocampus). The cerebral cortex and hippocampus in insulin-resistant rats on a normal diet exhibited neurons with pyknotic nuclei, perineuronal edema, red neurons, and areas of gliosis with evidence of Rosenthal fibers—75% of the brain tissue, grade 3. However, IR + KD showed neuronal injury: red neurons, tissue edema, and gliosis—25% of the brain tissue (grade 2 for hippocampus and cerebral cortex). In contrast, IR + ND + nifedipine rats displayed neuronal damage, incsluding neurons with pyknotic nuclei and perineuronal edema, red neurons, and regions of gliosis with evidence of Rosenthal fibers for the cerebral cortex, but the hippocampus showed no gliosis and similar features as the cerebral cortex—45% of brain tissue (grade 3, 15% of the brain tissues, and grade 2 scoring for cerebral cortex and hippocampus, respectively). IR rats + KD + nifedipine demonstrated neuronal injury: neurons with pyknotic nuclei, perineuronal edema, and red neurons—10% of the brain tissue (grade 1 for both cerebral cortex and hippocampus).

### 2.9. Nifedipine and KD Noticeably Mitigated Aβ and Tau Protein Levels

The hippocampal and cerebral cortex Aβ and tau protein levels were significantly increased in all groups as compared to the normal rat group. Meanwhile, the KD administration significantly decreased their levels compared to the IR + ND group. Notably, IR + ND + nifedipine treatment markedly increased the Aβ compared to the IR + KD group in both tissues of the study compared to the IR + KD group. Meanwhile, tau protein levels were significantly increased in the cerebral cortex and significantly decreased in the hippocampus compared to the IR + KD group. The best results were found in the IR + KD + nifedipine group as there was a significant decrease in Aβ levels as compared to the diseased groups, with an associated marked reduction in the tau protein levels compared to both groups fed an ND in cerebral cortex, but in the hippocampus both Aβ and tau protein levels were decreased markedly as compared to all diseased groups (Figure 8).

In conclusion, the results of this experiment can be summarized in the following Figure 9. Appendix A. is available and can be downloaded.

## 3. Discussion

This study sheds light on how different diets and nifedipine, the drug of study, affect the body in different aspects. Due to growing awareness about drugs’ safety and limitations, a non-negligible population has tried a KD to reduce body weight regardless of its negative effects on the body [13]. On the other hand, ignorant people also misuse drugs by using them to alleviate medical conditions or metabolic syndrome which may not correlate directly to their primary pharmacologic effect.

Overconsumption of fructose, used by many people as a sweetening agent instead of sucrose, for a long period, induces insulin resistance [14]. Just changing the type of diet from a high-fructose diet to a normal diet including moderate carbohydrates, as applied in the current study, might help to reverse insulin resistance and improve lipid profiles [15,16].

Following a balanced diet that can dramatically reduce body weight. However, the KD failed to alleviate body weight, insulin resistance, and lipid profile, as found by following the ND, as the KD is based mainly on fats as a source of energy, and the failure of skeletal muscle to oxidize lipids results in fat accumulation [17]. The imbalanced KD pattern creates a stressful state in the body leading to the release of high levels of cortisol through the activation of the hypothalamic–pituitary–adrenal axis (HPA) in response to low levels of carbohydrates [15]. Undoubtedly, high levels of cortisol associated with systemic metabolic disorders may be represented by insulin resistance, hyperinsulinemia, hyperglycemia, high levels of cholesterol, LDL, and TGs and accompanied by low levels of HDL. Moreover, there may be false positive results in reducing weight gain. To put it another way, the percentage of decreasing body weight was illusory and low compared to just having a balanced diet with a moderate amount of carbohydrates to meet body needs [18]. However several studies in the last few years, such as a study by Paoli, et al., have suggested the improvement of insulin resistance by a KD, based on their findings [19,20]. Also, Yilmaz, et al.’s study has a different point of view on the ability of a KD to improve lipid profile records [21].

Nifedipine treatment has been shown to impact energy utilization, leading to a decrease in body weight. This is due to its ability to increase the expression of peroxisome proliferator-activated receptor-γ co-activator-1α (PGC-1α) in skeletal muscle. Additionally, it promotes oxygen usage resulting in increased lipid oxidation and energy production [22]. However, its effect on metabolic disorder was disappointing. While nifedipine can reduce atherosclerotic cholesterol lesions [23], it did not significantly affect serum lipid levels, as found in this study.

Similarly, its effect on HOMA-IR was frustrating, as nifedipine failed to improve glucose or even insulin levels. This finding aligns with those of Naidu et. al., who found that nifedipine induced hyperglycemia through a postulated inhibition mechanism of baseline and glucose-induced insulin production [24]. Ca^2+^ plays a vital role in pancreatic β-cells, in both cell survival and insulin release. Glucose triggers ATP-dependent potassium channels that induce membrane depolarization, and voltage-gated l-type Ca^2+^ channels are turned on to induce the liberation of intracellular Ca^2+^ from ER stores which triggers insulin release. We can conclude that nifedipine may be effective in diabetic patients but not in insulin-resistant ones [25]. However, Tsukuda et al. and Lwai et al. have different points of view. They found that nifedipine can improve insulin resistance by increasing the activation of peroxisome proliferator-activated receptor-γ (PPAR-γ) [26,27].

In line with this study, Kackley et al. reported that the KD failed to restore normal levels of BDNF and hypothesized that increasing BDNF depends on creating a certain level of ketosis [28]. However, Evans et. al. and Stubbs et al. reported that after body ketosis adaptation, during a KD, the BDNF level increased [29,30]. Notably, nifedipine decreases BDNF levels, as Ca^2+^ plays an important role in the BDNF expression pathway [31].

GSK3β regulates insulin signaling in the metabolism process of glucose in different organs [32]. Insulin and Aβ share the same enzyme in their central metabolic process [33]. Although the ND alleviated insulin resistance and improved insulin sensitivity by normalizing IDE and GSK3β levels, with the resulting improvement of lipid profile peripherally, the ND could not resolve hyperphosphorylated tau protein or Aβ accumulation which was reflected by the defective cognition as seen in the MWM test.

The KD administration has improved long-term but not short-term memory, as measured in the MWM test. Although the KD has been shown to reduce the high levels of Aβ and tau protein, which indicates an improvement in cognitive function, this could be attributed to increasing the synthesis of new mitochondria, boosting ATP production, and reducing the production of reactive oxygen species (ROS) compared to the high level of ROS produced by glucose metabolism. It also helps decrease the import of amyloid precursor protein (APP) into the mitochondria and modify gene expression linked to neurodegenerative diseases [34]. The KD may be responsible for short-term memory issues, as it can create a stressful condition in the body by raising cortisol levels [15] leading to distraction and lack of concentration, as distinguished from memory. Meanwhile, the KD did not normalize IDE or GSK3β levels, as insulin resistance remained untreated.

Insulin resistance, not treated in both groups given nifedipine, was associated with high levels of GSK3β and low levels of IDE. However, administration of the KD has been shown to modify the effect of nifedipine by reducing the levels of Aβ and tau, while also decreasing GSK3β levels. It is known that the accumulation of Ca^2+^ in the cytosol of neural cells may result in the accumulation of Aβ and tau proteins [35]. Therefore, it can be concluded that nifedipine alone possesses the potential to reduce the accumulation of these proteins.

## 4. Material and Methods

### 4.1. Animals

In this investigation, adult male albino Wistar rats (130–180 g) [36] were housed, three rats per cage, at 28 ± 2 °C in plastic cages under a normal light/dark cycle with free access to food and water for fourteen days of acclimatization. The Ethical Committee for Animal Experimentation at the Faculty of Pharmacy, Cairo University approved this study (Permit Number: PT-(3046)).

### 4.2. Chemicals and Drugs

Fructose was purchased from the Safety Company (Cairo, Egypt). EIPICO (Cairo, Egypt) kindly gifted nifedipine that was dissolved in polyethylene glycol (PEG) 400 (43 mg of nifedipine was dissolved in 10 mL of PEG). The KD was designed to meet the nutritional requirements of adult rats as recommended by the American Institute of Nutrition (AIN-93M): protein (casein: 142.09; L-cysteine: 4.887), fats (soybean oil: 114.03; lard: 187.8; butter: 406), and carbohydrates (dextrin only: 30) [37]. The chemicals utilized during this experiment were research-grade.

### 4.3. Induction of Insulin Resistance (IR) in Rats

Fructose (10%) was administered to male Wistar albino rats [36] for 8 successive weeks for the induction of insulin resistance. Only rats suffering from insulin resistance, assessed by the oral glucose tolerance test (OGTT), were selected to complete the study.

After 8 weeks of receiving 10% fructose treatment, rats were fasted overnight to measure the FBG levels. After 12 h, blood glucose levels were recorded at different time intervals of 0, 30, 60, and 90 min after oral glucose administration (2.5 mg/kg; p.o) [38]. An ACCU-check^®^ Performa glucometer and test strips were used to detect the blood glucose levels.

### 4.4. Experimental Design: (Experimental Code: PO341)

Thirty male albino Wistar rats were randomly divided into five groups (six rats each) [39,40] in order to use the minimum number of animals which can achieve the required statistics according to animal ethics, as follows:Group 1 (normal control rats): this group was administered nifedipine vehicle (PEG 400) p.o. for 5 weeks (9th week till the end of the 14th week).Group 2 (IR + ND): rats received 10% fructose for 8 weeks to induce insulin resistance, then were converted to ND for an additional 5 weeks.Group 3 (IR + KD): insulin-resistant rats were fed KD for an additional 5 weeks [41].Group 4 (IR + ND + nifedipine): insulin-resistant rats fed with ND were concurrently treated with nifedipine (5.2 mg/kg, p.o.) for 5 weeks [42].Group 5 (IR + KD + nifedipine): insulin-resistant rats fed with KD were concurrently treated with nifedipine (5.2 mg/kg, p.o.) for 5 weeks.

The animals’ body weights were measured daily, and the net percentage increase or decrease in body weights was calculated. The MWM behavioral test was conducted twenty-four hours after the last nifedipine dose. Then, animals were sacrificed on the 92nd day by decapitation under anesthesia (2.5% thiopental sodium, 30 mg/kg; i.p.) [43]. Brains were dissected, and hippocampi were isolated and used for assessment of biochemical parameters. Tissue analysis in histological and immunohistochemical examinations was carried out using both the cerebral cortex and hippocampi. Bodies of dead rats were frozen at −80 °C till incineration following the procedures of the ethical committee, Faculty of Pharmacy, Cairo University.

### 4.5. Behavioral Test: Morris Water Maze (MWM) Test

The MWM test is a very valid test and the most widely used behavioral test in recent and older research. After the four weeks of treatment, the MWM test was used to evaluate cognitive performance on the 84th day of the trial. The test was carried out over five days. It was performed in a 150 cm diameter and 50 cm high circular pool with a white inner wall. The rounded pool was divided into four sections: southeast, southwest, northeast, and northwest. The rounded platform, which was set once in each quadrant, was constructed with specific dimensions: 30 cm in height and 15 cm in diameter. The rounded pool was filled with water and colored with non-toxic ink to hide the platform 1 cm below the water surface. The temperature was kept at 22–25 °C. Throughout the test, fixed items were placed against the pool’s walls to aid the animals’ spatial awareness. There are two phases to the MWM: the acquisition phase (first phase) and the probing phase (second phase) [44,45].

#### 4.5.1. Acquisition Phase (Short-Term Memory Test)

Each rat was allowed 120 s in each trial to escape from the water using the concealed platform during the first phase (acquisition phase), conducted over four consecutive days. In each trial, the time spent to reach the platform was documented. The rat was allowed to remain on the platform for only ten seconds. If the rat did not reach the platform within 120 s, the rat was directed to reach it, then, the time was recorded as 120 s. Each rat was removed and dried after each experiment.

#### 4.5.2. Probe Phase (Long-Term Memory Test)

The platform was removed on the 5th day, and rats were allowed to swim for 60 s only. The time spent in the target quadrant was recorded for all.

### 4.6. Biochemical Assessment

After performing the MWM test, rats were fasted overnight. Blood samples were collected from retro-orbital plexuses via micro-hematocrit heparinized capillary tubes and were left for 20 min before centrifugation at 4000 rpm for 15 min to isolate the sera. Sera were immediately kept at −80 °C till the biochemical assays [36,46].

#### 4.6.1. Homeostatic Model Assessment of Insulin Resistance (HOMA-IR)

FBG was evaluated in serum using a glucose colorimetric method (cat. No. GAGO-20; Sigma-Alrich, Saint Louis, MO, USA). Fasting insulin levels (FINS) were measured in serum using the rat insulin enzyme-linked immunosorbent assay (ELISA) kit (Cat. No. MBS281388; MyBioSource, San Diego, CA, USA). HOMA-IR was calculated as follows: [FBG (mg/dL) × FINS (μIU/mL)/405] [12]. 

#### 4.6.2. Lipid Profile Measurements

Following a colorimetric method, cholesterol, HDL and LDL, and TGs were measured in serum using the following kits: Cat. No. Z5030055; BioChain, Hayward, CA, USA, Cat. No. Z5030057; BioChain, Hayward, CA, USA, Cat. No. 5603-01; XpressBio, Frederick, MD, USA, respectively. 

#### 4.6.3. ELISA Technique

Via an ELISA technique, IDE, BDNF, and GSK3β were detected in tissue homogenates of hippocampus by using an ELISA reader (TECAE, A 5082) and the following ELISA kits: IDE (Cat. No. MBS722683; MyBioSource, San Diego, CA, USA), BDNF (Cat. No. MBS355345; MyBioSource, San Diego, CA, USA), and GSK3β (Cat. No. MBS909078; MyBioSource, San Diego, CA, USA). Procedures were carried out according to the manufacturers’ instructions.

### 4.7. Histopathological Investigation

The hippocampus and cerebral cortex were used for histopathological examination. Tissues were fixed for 48 h in 10% neutral buffered formalin, then were embedded in paraffin blocks and dehydrated. For histological and immunohistochemical studies, tissues were cut into very thin sections (5 μm in thickness) by using a rotatory microtome. Sections were fixed on MASGP-coated slides (Matsunami Glass Ind., Osaka, Japan) and were stained with hematoxylin–eosin (H&E) stain [36,46].

### 4.8. Immunohistochemical Analysis

For the immunohistochemical study, sections from the selected paraffin blocks were cut into 4-micrometer thick sections for immunohistochemical (IHC) staining. Slides were prepared and incubated with primary anti-β amyloid antibody (Cat. No. A17911; ABclonal, Woburn, MA, USA) and anti-tau antibody (Cat. No. A1103; ABclonal, Woburn, MA, USA). This was followed by incubations with the appropriate secondary antibody (PI 0207, Rev. G DCN: 3129; Bio SB, Santa Barbara, CA, USA). All slides were lightly counterstained with hematoxylin for 30 s before dehydration and mounting.

Neurons with cytoplasmic reaction to Aβ and tau were considered positive, while extracellular deposits with reaction to β-amyloid were considered positive. Semi-quantitative analysis of stained tissue sections was performed through modified Allred scoring system guidelines [47]. The number of positive neurons and the percentage of extracellular deposits in 3 hpf (400×) for the cerebral cortex and (200×) for the hippocampus were measured using ImageJ software v1.54g [48].

### 4.9. Statistical Analysis

Statistical analysis of all results was reported as median and interquartile range, either min. to max. or 25% to 75% (*n* = 6) [39,40]. Statistical significances were assessed with one-way analysis of variance (ANOVA) followed by the Tukey–Kramer test for post hoc comparisons between groups, except for the MWM test which was assessed using two-way ANOVA. Statistical analysis was performed using the computer program GraphPad Prism. The acceptable level for statistical significance was *p*, 0.05.

## 5. Conclusions

The study showed that nifedipine and the KD had ameliorative effects on the hippocampal histopathological changes, consistent with the improved immunohistochemical results of Aβ and tau levels. In conclusion, this study confirmed the ability of the KD to relieve accumulated Aβ and tau in neurons. However, it is important to note that the KD should only be followed for a limited time as prolonged use can have negative effects on metabolic syndrome and put the body under stress due to low carbohydrate intake. This can lead to the release of cortisol and the appearance of anxiety disorders. So, it is advised to follow a moderate and balanced diet instead of a stressful one. Interestingly, the best results in treating insulin-resistance-induced cognitive dysfunction were achieved when nifedipine was prescribed alongside the KD without a clear mechanism, peripherally and centrally. It is recommended to investigate the exact mechanism of nifedipine by which it modulated the KD effect in future studies. Based on Ca^2+^’s role in insulin release, we can conclude that nifedipine may be effective in diabetic patients, as hyperglycemia increases the intracellular level of Ca^2+^. Insulin-resistant individuals have normal levels of glucose and, hence, normal intracellular levels of Ca^2+^. So, nifedipine cannot provide a real improvement concerning insulin resistance.

Further studies are needed to determine if cardiovascular disease patients would benefit from following a KD while also receiving nifedipine. We may recommend another experiment studying the long-term effect of nifedipine.

One of the limitations of this study is that the MWM test was only used for assessing cognitive behaviors. In addition to the MWM test, other behavioral tests, e.g., open field habituation test, Y-maze test, and 8-arm radial maze test, can be performed in upcoming experiments. Another limitation was the limited number of animals in each experimental group (*n* = 6), hence future studies on the same topic should include a higher number of animals to increase the validity of the results.

## Figures and Tables

**Figure 1 pharmaceuticals-17-01054-f001:**
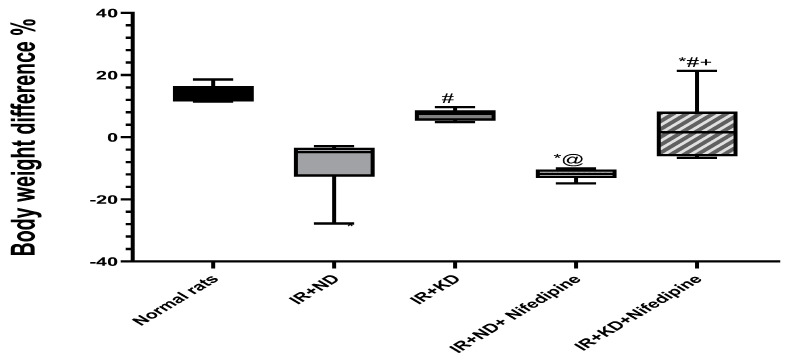
Effect of the normal diet (ND) and the ketogenic diet (KD) with or without nifedipine (5.2 mg/kg/day, p.o.) on the body weight difference, Final weight of IR at the end of the experiment— their weights after 8 weeks’ treatment with 10% fructose in insulin-resistant (IR) rats. Data are represented as median and interquartile percent (25–75%) and analyzed using one-way ANOVA followed by Tukey–Kramer test, with * *p* ≤ 0.05 vs. normal rats; # *p* ≤ 0.05 vs. IR + ND; @ *p* ≤ 0.05 vs. IR + ketogenic diet; and + *p* ≤ 0.05 vs. IR + ND + nifedipine group.

**Figure 2 pharmaceuticals-17-01054-f002:**
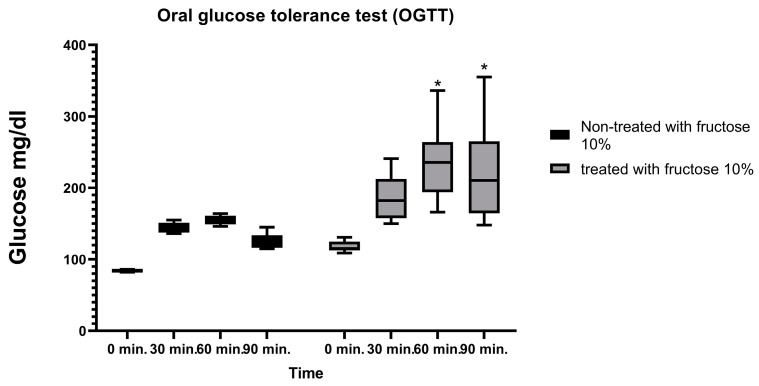
Effect of 10% fructose treatment after 8 weeks on the oral glucose tolerance test (OGTT). Blood glucose levels were evaluated before glucose administration and after administration of 2.5 mg/kg glucose at 30, 60, and 90 min. Results are represented as the median and interquartile range (min. to max.) (*n* = 6) and analyzed using two-way ANOVA followed by the Bonferroni test, with * *p* ≤ 0.05 vs. normal rats.

**Figure 3 pharmaceuticals-17-01054-f003:**
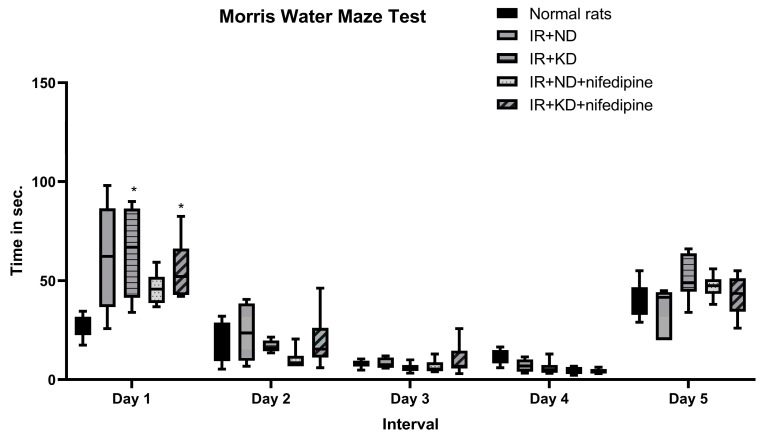
Effect of the ND and KD with or without nifedipine (5.2 mg/kg/day, p.o.) on the behavioral response in IR rats. Results are represented as median and interquartile range (min. to max.) (*n* = 6) and analyzed using two-way ANOVA followed by Tukey–Kramer test, with * *p* ≤ 0.05 vs. normal rats.

**Figure 4 pharmaceuticals-17-01054-f004:**
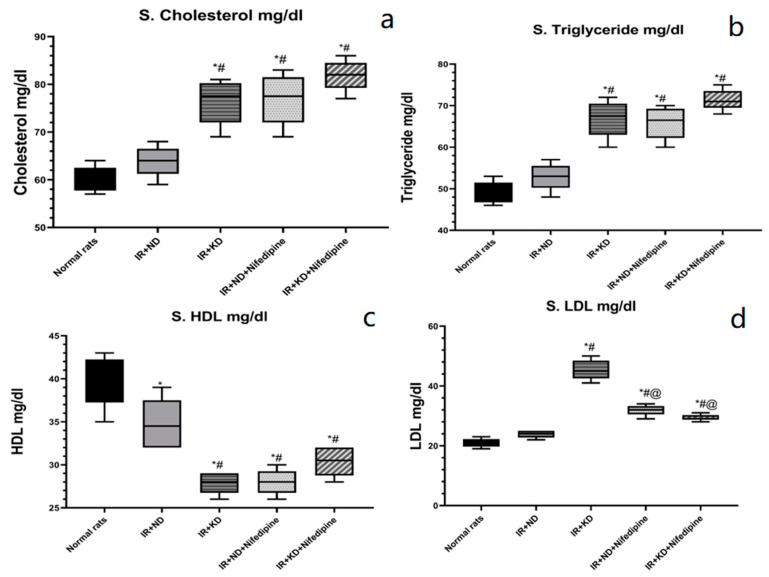
Effect of the ND and KD with or without nifedipine (5.2 mg/kg/day, p.o.) on serum lipid profile: (**a**) Cholesterol, (**b**) triglycerides, (**c**) high-density lipoprotein (HDL), and (**d**) low-density lipoprotein (LDL) (**d**) in (IR) rats. Results are represented as median and interquartile range (min. to max. with all points) (*n* = 6) and analyzed using one-way ANOVA followed by Tukey–Kramer test, with * *p* ≤ 0.05 vs. normal rats; # *p* ≤ 0.05 vs. IR + ND; and @ *p* ≤ 0.05 vs. IR + ketogenic diet.

**Figure 5 pharmaceuticals-17-01054-f005:**
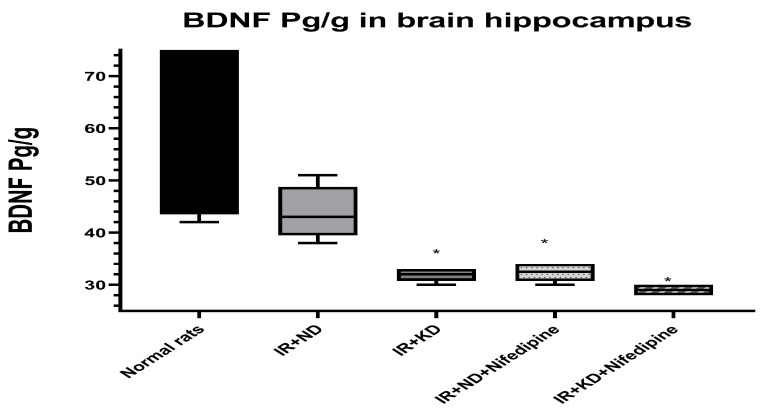
Effect of the ND and KD with or without nifedipine (5.2 mg/kg/day, p.o.) on the brain-derived neurotrophic factor (BDNF) in the brain hippocampus of IR rats. Results are represented as median and interquartile range (25–75%) (*n* = 6) and analyzed using one-way ANOVA followed by Tukey–Kramer test, with * *p* ≤ 0.05 vs. normal rats.

**Figure 6 pharmaceuticals-17-01054-f006:**
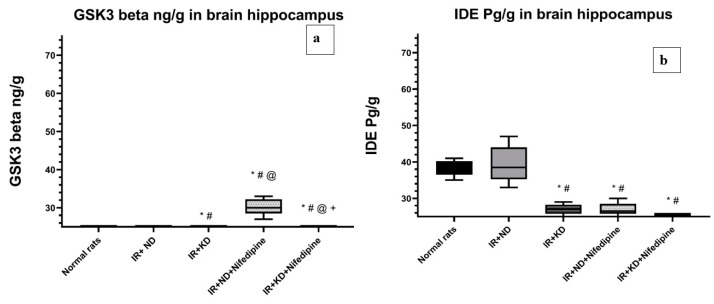
Effect of the ND and KD with or without nifedipine (5.2 mg/kg/day, p.o.) on (**a**) glycogen synthase kinase-3 beta (GSK3β) and (**b**) insulin-degrading enzyme (IDE) in the brain hippocampus of IR rats. Results are represented as median and interquartile range (25–75%) (*n* = 6) and analyzed using one-way ANOVA followed by the Tukey–Kramer test, with * *p* ≤ 0.05 vs. normal rats; # *p* ≤ 0.05 vs. IR + ND; @ *p* ≤ 0.05 vs. IR + ketogenic diet; and + *p* ≤ 0.05 vs. IR + ND + nifedipine group.

**Figure 7 pharmaceuticals-17-01054-f007:**
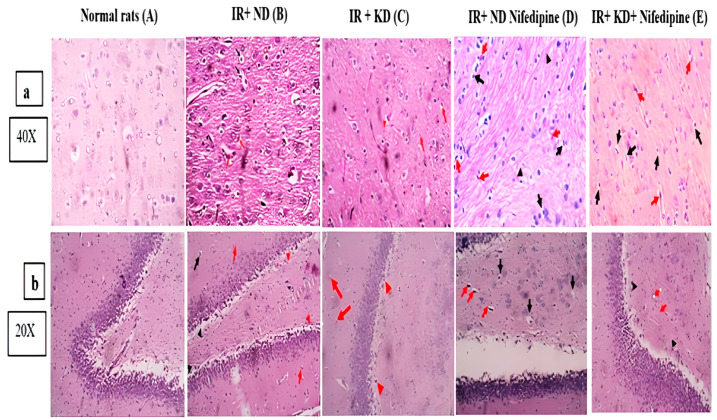
Effect of the ND and KD with or without nifedipine (5.2 mg/kg/day, p.o.) on histopathological examinations of insulin-resistant rats. Photomicrographs of hematoxylin-and-eosin-stained (**a**): for cerebral cortex and (**b**): for hippocampus slices from various groups. (**A**) Brain tissue that is uniform and without any neuronal damage. Grade 0 for both tissues. (**B**) The perineuronal edema and neurons with pyknotic nuclei (black arrows), Rosenthal fibers may be seen in gliosis (black arrowheads) regions, and the red arrows represent red neurons for both cerebral cortex and hippocampus, 75% of brain tissue. Grade 3 for both tissues. (**C**) Red arrows indicate that there are a few scattered red neurons, for both the cerebral cortex and hippocampus. For the cerebral cortex only, tissue edema (red arrowheads) is seen. The red neurons are sparsely distributed (red arrow). Perineuronal edema (black arrows), tissue edema (red arrowheads), and gliosis (black arrowheads) are all seen, 25% of brain tissue. Grade 2 for both tissues. (**D**) The red neurons are sparsely distributed (red arrow). Perineuronal edema (black arrows), tissue edema (red arrowheads), and gliosis in cerebral cortex only (black arrowheads) are all seen in both tissues, 45% of brain tissue. Grade 3 for cerebral cortex and 15% of brain tissue for hippocampus. Grade 2. (**E**) Neurons in cerebral cortex and hippocampus with pyknotic nuclei and perineuronal edema are seen (black arrows), 10% of brain tissue. Grade 1 for both tissues. There are gliosis regions with Rosenthal fibers in cerebral cortex only (black arrowheads). There are red neurons (shown by red arrows). Pictures captured at magnification power ×40 for cerebral cortex and ×20 for hippocampus.

**Figure 8 pharmaceuticals-17-01054-f008:**
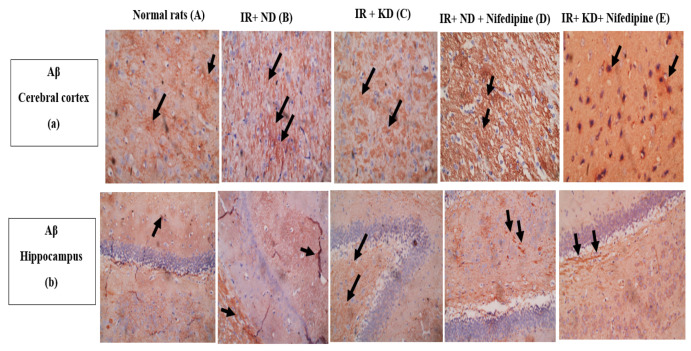
(**a**,**b**) (**A**) Weak focal levels of Aβ (black arrows). (**B**) Significant increase in levels of Aβ (black arrows). (**C**) Significant reduction in levels of Aβ (black arrows). (**D**) Significant increase in levels of Aβ (black arrows). (**E**) Weak focal levels of Aβ (black arrows). Pictures captured at magnification power ×40 for cerebral cortex and 20× for hippocampus. (**c**,**d**) (**A**) Weak levels of tau in the cytoplasm of a few neuron cells (black arrows). (**B**) There is an increase in the levels of tau in many neurons (black arrows). (**C**) There is a reduction in the levels of tau in many neurons (black arrows). (**D**) There is an increase in the levels of tau in many neurons (black arrows). (**E**) There are weak levels of tau in cytoplasm of some neurons (black arrows) in cerebral cortex while a significant increase in the levels of tau in cytoplasm of some neurons (black arrows). Pictures captured at magnification power ×40 for cerebral cortex and 20× for hippocampus. Statistical analysis showed the effect of the ND and KD with or without nifedipine (5.2 mg/kg/day, p.o.) on immunohistochemical staining of cerebral cortex and hippocampal levels of amyloid β (Aβ) (**e**,**f**) and tau protein (**g**,**h**) of insulin-resistant rats. Results are represented as median and interquartile range (25–75%) (*n* = 6) and analyzed by one-way ANOVA followed by Tukey–Kramer test, with * *p* ≤ 0.05 vs. normal rats; # *p* ≤ 0.05 vs. IR + ND; @ *p* ≤ 0.05 vs. IR + ketogenic diet; and + *p* ≤ 0.05 vs. IR + ND + nifedipine group.

**Figure 9 pharmaceuticals-17-01054-f009:**
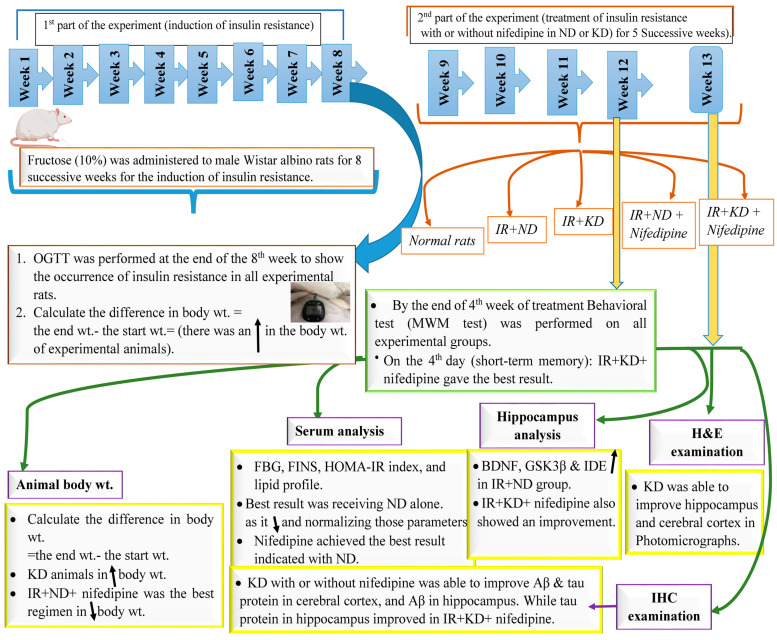
A schematic presentation for the results of the study.

**Table 1 pharmaceuticals-17-01054-t001:** Effect of the normal diet (ND) and ketogenic diet (KD) with or without nifedipine (5.2 mg/kg/day, p.o.) on the weight gain of insulin-resistant (IR) rats.

Experimental Groups	Mean Delta Body Weight after 8 Weeks of Treatment with 10% Fructose	Percentage %	Mean Delta Final Body Weight after Nifedipine and Nutrition Treatment	Percentage %
Normal rats	94	87.29%	30.34	14.51%
IR + ND	217.5	138.54%	−25.5	−6.07%
IR + KD	198	111.86%	17.5	4.58%
IR + ND + Nifedipine	243.5	148.48%	−46	−10.76%
IR + KD + Nifedipine	206.5	130.7%	32.5	8.86%

**Table 2 pharmaceuticals-17-01054-t002:** Effect of the normal diet (ND) and ketogenic diet (KD) with or without nifedipine (5.2 mg/kg/day, p.o.) on the fasting blood glucose (FBG), fasting insulin (FINS) levels, and homeostatic model assessment of insulin resistance (HOMA-IR) index of the insulin-resistant rats.

Experimental Groups	FBG			FINS			HOMA-IR		
	Median	25% Percentile	75% Percentile	Median	25% Percentile	75% Percentile	Median	25% Percentile	75% Percentile
Normal rats	70	68	71.75	7.6	7.35	8.13	1.34	1.23	1.48
IR + ND	76	73.25	83.50	8.4	8	9.33	1.59	1.35	1.84
IR + KD	86 *	78.25	90.75	10.30 *#	9.53	11.73	2.18 *#	1.77	2.79
IR + ND + Nifedipine	85.50 *	79	91.75	9.65 *	8.88	10.15	2.05 *	1.67	2.29
IR + KD + Nifedipine	93.50 *#	89	96.75	10.30 *#	9.78	10.90	2.38 *#	2.14	2.60

Results are represented as median and interquartile range (min. to max.) with * *p* ≤ 0.05 vs. normal rats; # *p* ≤ 0.05 vs. IR + ND. ND: normal diet, KD: ketogenic diet.

## Data Availability

The raw data supporting the conclusions of this article will be made available by the authors on request.

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
