# Peer review of "Nifedipine Improves the Ketogenic Diet Effect on Insulin-Resistance-Induced Cognitive Dysfunction in Rats"

_pharmaceuticals, 2024, doi:10.3390/ph17081054_

Round 1

Reviewer 1 Report

Comments and Suggestions for Authors

The study by Abdel-Kareem et al. investigates the effects of nifedipine, a calcium channel blocker, in conjunction with normal diet or ketogenic diet on cognitive dysfunction induced by insulin resistance in rats. Insulin resistance was induced using high fructose consumption, and various biochemical markers including insulin sensitivity, lipid profile, and cognitive performance were assessed following treatment with nifedipine and dietary interventions. I have to mention the strength of the work - the study employs a good experimental design involving induction of insulin resistance through fructose consumption followed by treatment with nifedipine alongside ND or KD. Data are presented clearly with appropriate statistical analysis, enhancing the clarity and reliability of the data.

1) However, while the study effectively measures outcomes, it lacks good mechanistic exploration of how nifedipine interacts with insulin signaling pathways or the specific metabolic effects of KD in this context.

2) The study uses a relatively small sample size (n=6), which may limit the generalizability of the findings. Increasing the sample size could enhance the robustness of the conclusions.

3) Consider additional behavioral tests or cognitive assessments to provide a more comprehensive evaluation of the effects on different aspects of cognitive function.

4) Investigate the long-term effects of nifedipine and KD on insulin resistance and cognitive function to understand sustainability and potential adverse effects over extended periods.

If the authors address these questions, the manuscript could be accepted after minor revisions. With a clearer mechanistic understanding and potentially larger sample sizes, this work could contribute to the field of neuropharmacology and metabolic disorders.

Reviewer 2 Report

Comments and Suggestions for Authors

This manuscript reports the beneficial effect in case of insulin resistance-induced cognitive deficit in a model of diabetic’s rats after treatment by a ketogenic diet.

The manuscript should be improved as stated in my comments.

Materials & Methods

Statistical analysis:

Data should not be presented as mean ± SEM. With this presentation, it is difficult to see the intra-variability.

Bar charts are not appropriate for continuous measures since they do not provide information about distribution of the data. Appropriate descriptive measures of the average and variability for continuous measures in tables, text, or graphs are the arithmetic mean and standard deviation if the data are sufficiently normally distributed, or the median and interquartile range [being the 25th and 75th percentile] if data are not sufficiently normally distributed, but not the standard error.

Thus, all data should be presented as boxplot with the median and interquartile range.

At least, Bar charts must be given with dots (one per animal).

Figures:

The graphic representation of the data in figures could be improved.

The figures legends should be revised: it is not necessary to repeat that rats received 10% fructose for 8 weeks…… (figure 3 to figure 8). This is (and must) be indicated in the Methods.

Figure 7: provide enlarged panels in the different photos to show gliosis.

Reference 38: all words are in capital letters (upper cases). Revise in lower cases.

Comments on the Quality of English Language

Reviewer 3 Report

Comments and Suggestions for Authors

Authors describe an interesting study about insulin resistance, ketogenic diet, and nifedipine. However, there are number of problems with presentation of the results and conclusions.

1. Abstract does not clearly distinguish between metabolic-related effects and brain-related effect. Overall, abstract is a confusing for a reader that has not read the paper.

2. Results need a schematic showing how many cohorts were used,  what treatments were applied, and when treatment where done. Granted that this information is described in the Methods, but it is a bit opaque. 

3. Figure 1 and Table 1 are poorly explained. I am not sure what those number are. I can guess, but the reader should not guess.

4. Figures 4, 5, and 6 do not specify in which tissue measurements were taken. I guess it is blood for the Fig. 4, but I am not so sure about the other two.

5. Figure 7 shows only representative images. You will need to show average grades for each conditions.

6. Figure 8: measurements are not properly explained, and language in confusing. For example, expression usually means mRNA levels, but it is obviously not the case.

Minor:

1. Plot individual data points instead of bar graphs.

2. Line 102-103: provide short explanation for the index.

3. Figure 6 is placed in the middle of the legend for Figure 6.

4. Figure 8 has 3 version of the legend

5. Supplemental Materials need to be better formatted and have short explanation for each data table.

Comments on the Quality of English Language

Some small problems with word usage like line 15 "suggested" is probably should be "prescribed", line 27 "continued" is perhaps "combined", and few others.

Round 2

Reviewer 2 Report

Comments and Suggestions for Authors

all comments have been taken into account and the manuscript has been well revised